# Unsupervised Functional Dependency Discovery for Data Preparation

**Zhihan Guo & Theodoros Rekatsinas**
Department of Computer Sciences
University of Wisconsin-Madison
Madison, WI 53715, USA
`{zhihan, thodrek}@cs.wisc.edu`

## Abstract

We study the problem of functional dependency(FD) discovery to impose domain knowledge for downstream data preparation tasks. We introduce a framework in which learning functional dependencies corresponds to solving a sparse regression problem. We show that our methods can scale to large data instances with millions of tuples and hundreds of attributes, while recovering true FDs across a diverse array of synthetic datasets, even in the presence of noisy data. Overall, our methods show an average $F_1$ improvement of $2\times$ against state-of-the-art FD discovery methods. Our system also obtains better $F_1$ in downstream data repairing task than manually defined FDs.

## 1 Introduction

Functional dependencies (FDs) are an integral part of data management systems. They are used in database normalization to reduce data redundancy and improve data integrity (Garcia-Molina et al., 1999). FDs are also critical in data preparation tasks, such as data profiling and data cleaning. For instance, FDs can help guide feature engineering in machine learning pipelines (Ghiringhelli et al., 2015) or can serve as a means to identify and repair erroneous values in the given dataset (Rekatsinas et al., 2017; Chu et al., 2013). Unfortunately, FDs are typically unknown and it requires significant effort and domain expertise to identify them.

Various works have focused on automating FD discovery, both in the database (Kruse & Naumann, 2018; Huhtala et al., 1999; Papenbrock et al., 2015) and the data mining communities (Mandros et al., 2017; Reimherr & L. Nicolae, 2013). The works in the database community study how to infer FDs that a dataset instance $D$ does not violate. These approaches are well-suited for database normalization purposes and for applications where strong closed-world assumptions on the given dataset $D$ hold. In contrast, the data mining community views FDs as statistical dependencies manifested in a dataset and has focused on information theoretic measures to estimate FDs. These approaches are more suited for data profiling and data cleaning applications. In this paper, we focus on FDs that correspond to statistical dependencies in the generating distribution of a given dataset.

**Challenges** Inferring FDs from data observations poses many challenges. First, the candidate space of possible FDs over a dataset increases exponentially with the number of attributes in a dataset. Many of the existing methods rely on pruning are shown to exhibit poor scalability as the number of columns increases (Kruse & Naumann, 2018; Mandros et al., 2017).

Moreover, in real-world datasets, missing or erroneous values introduce uncertainty in FD discovery. This poses a challenge as noise can lead to the discovery of spurious FDs or to low recall with respect to the true FDs in a dataset. The performance of existing methods (Kruse & Naumann, 2018; Mandros et al., 2017), in terms of runtime and accuracy, is sensitive to factors such as sample sizes, prior assumptions on error rates, and the amount of records available in the input dataset. This makes these methods cumbersome to use in data preparation tasks with heterogeneous datasets.

**Our Contributions** We propose a framework that relies on *structure learning* to solve FD discovery. Specifically, we show that discovering FDs is equivalent to learning the graph structure over

binary random variables. *A key result in our work is to model the distribution that FDs impose over pairs of records instead of the joint distribution over the attribute-values of the input dataset.*

We compare our method against state-of-the-art methods from both the database and data mining literature over a diverse array of synthetic datasets with varying number of attributes, domain sizes, records, and amount of errors. We find that our method scales to large data instances with hundreds of attributes and yields an average $F_1$ improvement in discovering true FDs of more than $2\times$ compared to competing methods.

We also examine the effectiveness of our system on downstream data preparation tasks as an alternative for manually defined domain knowledge. We show that dependencies discovered via our method lead to higher-quality repairs in data repairing tasks compared with manually specified dependencies. This demonstrates that our FD discovery method offers a viable solution to automating weakly supervised data preparation tasks.

**Outline**   In Section 2, we introduce our probabilistic model for FD discovery and the structure learning method we use to infer its graphical structure. In Section 3, we present an experimental evaluation of our system, and conclude in Section 4.

## 2   FRAMEWORK

In this section, we formalize the problem of functional dependency discovery and provide an overview of our solution.

### 2.1   PROBLEM STATEMENT

We consider a relational schema $R$ associated with a probability distribution $P_R$. We assume access to a noisy dataset $D'$ that follows schema $R$. Given a noisy data instance $D'$, our goal is to identify the FDs that characterize the distribution $P_R$ that generated the clean version of $D$.

In our work, we combine the probability-based and logic-based interpretations of FDs. For any pair of tuples $t_i$ and $t_j$ sampled from $P_R$, we denote $I_{ij} = \mathbb{1}(t_i[Y] = t_j[Y])$ where $\mathbb{1}(\cdot)$ is the indicator function, and denote $t_i[\mathbf{X}]$ the value assignment for attributes $\mathbf{X}$ in tuple $t_i$. Given a distribution $P_R$, we say that an FD $\mathbf{X} \to Y$, with $\mathbf{X} \subseteq R$ and $Y \in R$, holds for $P_R$ if for all pairs of tuples $t_i, t_j$ in $R$ we have that

$$\Pr(I_{ij} = 1; t_i[\mathbf{X}], t_j[\mathbf{X}]) \propto \begin{cases} 1, & \text{when } t_i[\mathbf{X}] = t_j[\mathbf{X}] \\ \theta, & \text{otherwise} \end{cases} \tag{1}$$

with $\theta = \sum_{y \in V(Y)} P_R(y; t_i[\mathbf{X}]) \cdot P_R(y; t_j[\mathbf{X}])$. This condition states that the two random events $\bigwedge_{A \in \mathbf{X}} t_i[A] = t_j[A]$ and $\mathbb{1}(t_i[Y] = t_j[Y])$ are deterministically correlated when the FD $\mathbf{X} \to Y$ holds, otherwise they are independent.

Under this interpretation, the problem of FD discovery corresponds to learning the structural dependencies amongst attributes of $R$ that satisfy the above condition. Specifically, our system is build upon a probabilistic graphical model, which consists of binary random variables that model these two random events. The edges in the model represent statistical dependencies that capture the relation in Equation 1. Each true FD in the data generating distribution corresponds to a directed subgraph with V-structure. Thus, we can reframe our goal as to learn the graphical structure of this probabilistic graphical model. Furthermore, this formulation enjoys better sample complexity than applying structure learning on the raw input dataset. We focus on the case of discrete random variables to explain this argument. The sample complexity of state-of-the-art structure learning algorithms is proportional to $k^4$ (Wu et al., 2018) where k is the size of the domain of a variable. Our model restricts the domain of the random variables to be k = 2, and hence, yields better sample complexity than applying structure learning directly on the raw input.

### 2.2   SOLUTION OVERVIEW

**Model Relaxation**   As learning the structure of a directed graphical model with V-structure patterns is NP-hard (Chickering et al., 2004). We relax our initial model to a *linear structural equation model* that approximates the condition in Equation 1. We use $Z_A \in \{0, 1\}$ $(A \in R)$ to denote the

random event of sampling two tuples from distribution $P_R$ such that they have the same value for attribute $A$. First, we relax the random variables $\{Z_A\}_{A \in R}$ to take values in $[0, 1]$ instead of $\{0, 1\}$. Second, from Equation 1, we have that when $\bigwedge_{A \in \mathbf{X}} t_i[A] = t_j[A] = \bigwedge_{A \in \mathbf{X}} Z_A = \texttt{True}$ it must be that $\mathbb{1}(t_i[Y] = t_j[Y]) = Z_Y = 1$. We consider the random vector $\mathbf{Z} = \{Z_{A_1}, Z_{A_2}, \ldots, Z_{A_{|R|}}\} \in [0, 1]^l$ that corresponds to the random variables associated with the attributes in schema $R$. Based on the aforementioned relaxed condition, FDs force this random vector to follow a *linear structured equation model*. Hence, we can write that:

$$\mathbf{Z} = B^T \mathbf{Z} + \epsilon, \tag{2}$$

where we assume that $E[\epsilon] = 0$ and $\epsilon_j \perp\!\!\!\perp (Z_{A_1}, \ldots, Z_{A_{j-1}})$ for all $j$, where $\perp\!\!\!\perp$ denotes conditional independence. Since our model corresponds to a directed graphical model, matrix $B$ is a strictly upper triangular matrix. $B$ is known as the *autoregression matrix* of the system (Loh & Bühlmann, 2014). For DAG $G$ with vertex set $V = \{Z_{A_1}, Z_{A_2}, \ldots, Z_{A_{|R|}}\}$ and edge set $E = \{(j, k) : B_{jk} \neq 0\}$, the joint distribution factorizes as $P(Z_{A_1}, \ldots, Z_{A_{|R|}}) = \prod_{j=1}^{|R|} P(Z_{A_j} | Z_{A_1}, \ldots, Z_{A_{j-1}})$. Given samples $\{\mathbf{Z}^i\}_{i=1}^N$, our goal is to infer the unknown matrix $B$.

**Structural Learning**   Our structure learning algorithm is built upon a recent result of Loh and Buehlmann (Loh & Bühlmann, 2014) on learning the structure of linear causal networks via inverse covariance estimation. Given a linear model as the one shown in Equation 2, it can be shown that the inverse covariance matrix $\Theta = \Sigma^{-1}$ of the model can be written as:

$$\Theta = \Sigma^{-1} = (I - B)\Omega^{-1}(I - B)^T \tag{3}$$

where $I$ is the identity matrix, $B$ is the autoregression matrix of the model, and $\Omega = \text{cov}[\epsilon]$ with $\text{cov}[\cdot]$ denoting the covariance matrix. The structure learning algorithm proceeds as follows: Suppose we have $N$ observations with an empirical covariance matrix $S$, we estimate sparse inverse covariance $\theta$ by solving the following optimization problem: $\min_{\Theta \succ 0} f(\Theta) := -\log \det(\Theta) + tr(S\Theta) + \lambda \|\Theta\|_1$ using *Graphical Lasso* (Friedman et al., 2008). Graphical Lasso is shown to scale favorably to large instances and hence is appropriate for our setting. Given the estimated inverse covariance matrix $\hat{\Theta}$, we use the Bunch-Kaufman algorithm to obtain a factorization of $\hat{\Theta}$ and obtain an estimate for the autoregression matrix $\hat{B}$.

**FD generation**   Finally, we use the autoregression matrix $\hat{B}$ to generate FDs. We do so by considering the non-zero off-diagonal entries of the estimated inverse covariance matrix. The final output of our model is a collection of discovered FDs of the form $\mathbf{X} \to Y$ where $\mathbf{X} \subseteq R$ and $Y \in R$.

## 3 EXPERIMENTS

We compare our system against several FD discovery methods on synthetic datasets and evaluate our system in a downstream data preparation task with real-world datasets.

**Using Our System to Discover Controlled FDs in Synthetic Datasets**   We generated synthetic datasets to capture different data properties and control data generating distribution. Then, We use them to compare our system against state-of-the-art methods over the goal of finding true FDs held for data generating distribution. We considered two competing methods: PYRO (Kruse & Naumann, 2018), the state-of-the-art FD discovery method in the database community, and RFI (Mandros et al., 2017), the state-of-the-art FD discovery approach in the data mining community. In the results, our system consistently outperforms all other methods in terms of $F_1$-score across all settings, with an $F_1$ improvement of more than $2\times$ on average. In detail, PYRO has an average precision of 0.24% and an average recall of 57.58% on datasets with low amount of noises ($\leq 1\%$). The behavior is expected as PYRO follows a logic-based interpretation of FDs. It aims to discover all FDs holding for a given dataset instead of the FDs imposed in the data generating distribution. For RFI, it exhibits poor scalability and only completed 9 out of 24 synthetic settings. For the cases RFI terminates, it exhibits high precision (an average of 84.44%) for small cardinality domains with a large number of samples and low amount of noises. As the sample size decreases or the noise rate increases we find that the performance of RFI drops significantly. Overall, the average precision and recall for

RFI are 24.41% and 64.81% on datasets with low amount of noises. Our system maintains good precision and recall for datasets with low amount of noises with an average precision of 85.60% and an average recall of 99.75%. *This verifies our hypothesis that structure learning along with the data transformation step introduced in Section 2.2 leads to a more accurate FD discovery solution.*

**Using Our System to Automate Data Cleaning**  Recent work (Rekatsinas et al., 2017) showed that integrity constraints such as FDs can be used to train machine learning models for data cleaning in a weakly supervised manner. A limitation of this work is that it relies on users to specify these constraints. Here, we test if our system can be used to automate this process and address this pain point. For our experiments, we use the open-source version of the system from (Rekatsinas et al., 2017), as it provides a collection of manually specified FDs for the Hospital dataset. We perform the following experiment:

(1) We compare the manual FDs in that repository with the FDs discovered by our system. The autoregression matrix output by our system is shown in Figure 1. We find that the discovered FDs are meaningful. For example, we see that attributes 'Provider Number' and 'Hospital Name' determine most other attributes. We also see that 'Address1' determines location-related attributes such as 'City', 'Zip code' and 'County'. In repairing, the precision, recall, and $F_1$ reported by the data cleaning system for the manual constraints are 0.91, 0.70, and 0.79 respectively, while the corresponding metrics for the FDs discovered by our system is 0.93, 0.72, and 0.81. We see that this performance is better than the manually specified FDs.

(2) We run PYRO and RFI respectively over the same dataset and evaluate the repairing performance with their discovered FDs. For RFI, it is more than $8000\times$ slower than PYRO and our system with the same hardware setting. From the discovered FDs, we find RFI has the problem of overfitting to the dataset – some discovered relations hold for the given dataset instance, but does not convey any real-world meaning. We attribute this behavior to the fact that the domain of the left hand side is too large compared with the domain of the right hand side. This makes it more likely to observe a spurious FD when the number of data samples is limited. In repairing, the precision, recall, and $F_1$ with RFI discovered FDs are 1.00, 0.61, 0.76 respectively. For PYRO, it outputs 434 FDs, many of which are not particularly meaningful. This results in a poor average precision of 0.16. These findings support the applicability and advantage of our system to discover FDs that are useful in downstream data preparation tasks.

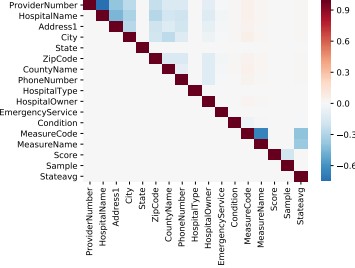

Figure 1: The autoregression matrix estimated by our system for the Hospital dataset.

## 4 CONCLUSIONS

We introduced our system, a structure learning framework to solve the problem of FD discovery for data preparation tasks. A key result in our work is to model the distribution that FDs impose over pairs of records instead of the joint distribution over the attribute-values of the input dataset. Specifically, we introduce a method that convert FD discovery to a structure learning problem over a linear structured equation model. We empirically show that our system outperforms state-of-the-art FD discovery methods and can produce meaningful FDs that are useful for downstream data preparation tasks.

## 5 ACKNOWLEDGEMENTS

This work was supported by Amazon under an ARA Award and by NSF under grant IIS-1755676.

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
