# OpenReview forum: "Unsupervised Functional Dependency Discovery for Data Preparation"
_ICLR.cc/2019/Workshop/LLD — LLD 2019_

### Official Review · AnonReviewer1 · 2019-03-31
**Elegant relaxation of functional dependency discovery as graph structure learning, validated by good empirical performance**

**Rating:** 4
**Confidence:** 2

**Review:**

Summary: The paper proposes a framework to relax the functional dependency (FD) discovery problem as a structure learning problem by focusing on the dependency between pairs of records. Graphical lasso is applied to obtain the sparse inverse covariance matrix, resulting in an approximate solution to the FD discovery problem. The proposed method is compared with prominent FD discovery methods such as PYRO and RFI, showing much higher precision and recall on a synthetic dataset. On a real dataset of data cleaning, the proposed method outperforms HoloClean, which uses manually written FDs.

Strengths:
1. The relaxation of the FD discovery problem to structure learning in elegant.
2. The empirical performance on synthetic and real datasets are impressive.

Weakness:
Lack of theoretical guarantee: perhaps there's a way to get error bound on B_hat from the error bound for Glasso?

---

### Official Review · AnonReviewer2 · 2019-04-07
**interesting approach to functional dependency discovery with empirical support**

**Rating:** 4
**Confidence:** 2

**Review:**

This paper introduces a structure learning framework for functional dependency (FD) discovery. The authors model the distribution of FDs over pairs of records by capturing dependencies of attribute-values in a graph structure.

The authors compare their approach to manually-specified dependencies and automated state-of-the-art methods from the database and data mining community.

I’ve included a list of strong points and points of confusion/questions below:
+ Figure 1 showing autoregression matrix shows incrementally verifies hypotheses in structure learning approach.
+ Empirical results support strong improvements over baselines.
- The authors describe the key result: “to model the distribution that FDs impose over pairs of records instead of the joint distribution over the attribute-values of the input dataset”. Could you further explore the theoretical/empirical improvement here? This would strengthen the motivation of the approach.
- What were the trade-offs considered in database/data mining communities-- what are the conceptual limitations from these communities that the authors were able to overcome?
- Additional datasets -- how does the method perform on other datasets with different error/data repairing characteristics (i.e. mentioned in the HoloClean paper, or others)?

The paper is well-written, and could be strengthened additional exploration to strengthen choice and comparison to baselines.The paper w ould be a reasonable fit for the workshop, under the challenges introduced via: “Representations to enforce structured prior knowledge (e.g. invariances, logic constraints).”

---

### Decision · Program_Chairs · 2019-04-08
**Acceptance Decision**

Accept